# Impact of perceived discrimination and coping strategies on well-being and mental health in newly-arrived migrants in Spain

Aina Gabarrell-Pascuet[1,2°], Amanda Lloret-Pineda[1°], Marta Franch-Roca[1], Blanca Mellor-Marsa[1], Maria del Carmen Alos-Belenguer[1], Yuelu He[1], Rachid El Hafi-Elmokhtari[1], Felipe Villalobos[3], Ivet Bayes-Marin[2,4], Lola Aparicio Pareja[5], Oscar Álvarez Bobo[5], Mercedes Espinal Cabezas[5], Yolanda Osorio[5,6], Josep Maria Haro[1,2], Paula Cristóbal-Narvaez[1,2]*

1 Teaching, Research & Innovation Unit, Parc Sanitari Sant Joan de Déu, Sant Boi de Llobregat, Barcelona, Spain, 2 Center for Biomedical Research on Mental Health (CIBERSAM), Madrid, Spain, 3 Fundació Institut Universitari per a la Recerca a l'Atenció Primària de Salut Jordi Gol i Gurina (IDIAPJGol), Barcelona, Spain, 4 Department of Medicine, School of Medicine and Health Sciences, Universitat Internacional de Catalunya, Sant Cugat del Vallés, Spain, 5 Servicio de Atención a la Migración en Salud Mental (SATMI), Parc Sanitari Sant Joan de Déu, Barcelona, Spain, 6 Equip Salut Mental Sense Sostre (ESMES), Parc Sanitari Sant Joan de Déu, Barcelona, Spain

° These authors contributed equally to this work.
* paula.cristobal@sjd.es

## Abstract

### Objectives

To explore how perceived discrimination impacts the emotional well-being and mental health of newly-arrived migrants in Spain; and to identify the coping strategies and behavioral changes used to deal with perceived discrimination.

### Design

102 individual audio-recorded in-depth qualitative interviews were conducted. The interviews were transcribed and analyzed through content analysis.

### Results

Negative emotions related to perceived discrimination included disgust, sadness, fear, loneliness, humiliation, sense of injustice, rage, feeling undervalued or vulnerable, and mixed emotions. Change in behaviors due to perceived discrimination comprised westernization or cultural assimilation, creating a good image, avoiding going out or leaving alone, hypervigilance, stop participating in politics, self-sufficiency, a positive adaptation, and paradoxically, becoming an oppressor. The identified coping strategies to deal with perceived discrimination were ignoring or not responding, isolation, self-medication, engagement in intellectual activities, leisure and sport, talking or insulting the oppressor, denouncement, physical fight or revenge, seeking comfort, increasing solidarity with others, crying, or using humor. Discrimination-related stress and related mental health problems were conveyed, as challenges related to substance abuse and addictive behaviors, mood, and anxiety.

**Data Availability Statement:** All relevant data are within the paper and its Supporting information files.

**Funding:** ALP, BMM, MCAB, YH, RHE, FV, and IBM work was endorsed by the project (II IN 200803 EN 162 FA 01), financed by the Spanish Ministry of Labor, Migrations and Social Security with the support of the European Commission (https://www.inclusion.gob.es/web/migraciones/fami). AGP work was financed by the Secretariat of Universities and Research of the Generalitat de Catalunya and the European Social Fund (2021 FI_B00839). PCN's work was supported by Sara Borell (CD20/00035) and financed by the Instituto de Salud Carlos III. The funders had no role in study design, data collection and analysis, publication decisions, or manuscript preparation.

**Competing interests:** The authors have declared that no competing interests exist.

## Conclusions

Findings establish initial evidence of the great impact of perceived discrimination on the health, emotional well-being, and behavior of newly-arrived migrants in Spain, alerting to the need for targeted policies and services to address the effects of discrimination in this population. Further research is needed to explore more closely the causes and effects of perceived discrimination on mental health, to develop more targeted and effective interventions.

## 1. Introduction

Nowadays, Spain is one of the leading destinations worldwide for receiving international migrants, although it was not established as a receiving country for migrations until the early 1980s [1]. In 2021, a total of 457,701 people arrived in Spain, reaching 5,440,148 foreigners living in Spain [2].

In recent years, international scientific evidence has pointed out the role of–structural, cultural, and individual–racism in health [3, 4]. More specifically, perceived discrimination has been conceptualized as a significant stressor affecting the mental health of the global migrant population [5, 6]. As a possible explanation, Sam & Berry [7] have extensively reported acculturation stress as adapting to the host country's cultural norms and values, while living with the country-of-origin standards. Most studies have associated those experiences of discrimination with significantly higher levels of mental health conditions such as anxiety, depression [8], substance abuse [9], or suicidal ideation [10]. Equivalently, perceived discrimination has shown correlations with depression and anxiety, and more strongly, with psychological distress [4, 6]. In terms of subjective well-being, Hadjar & Backes [11] found a great disadvantage on first-generation migrants. Paradoxically, other authors concluded that life satisfaction of the migrant population is generally positive, depending mainly on the cultural origin–with higher levels of well-being associated to closer cultures–and gender. Moreover, the perception of stigma and discrimination hinders mental health services; it favors unmet mental health needs [12] and legal and language barriers in accessing healthcare or primary care services [13]. This is supported by empirical evidence showing how sociodemographic and socioeconomic disadvantage [i.e., gender, race, socioeconomic status, and age] impact health. Still, these factors alone cannot explain inequities in the migrant populations' health [14].

Despite the extensive body of research supporting a higher prevalence of mental health problems among migrant communities, few studies have focused on the types of perceived discrimination and subjectively perceived rejection experiences in the Spanish context. Besides, there is a lack of studies regarding the impact of these incidents on migrants' mental health and emotions. In addition, scarce literature examines the associated behavioral changes and coping strategies. In the Spanish territory, migrant communities experience a greater burden of mental health disorders, perceived discrimination, and negative feelings like rejection [15], compared to those born in Spain [16]. According to Gil-González et al. [17], discrimination may constitute a risk factor for health in migrant workers and could explain some health inequalities among migrant populations in the Spanish society. Previous research has highlighted that structural stigma and minority stress mechanisms can encourage the deterioration of the mental health of migrants in the host country [12, 14, 18]. Additionally, a high percentage of the migrant sample interviewed by Agudelo-Suárez et al. [19] reported perceived discrimination, associated mainly with their condition of being a migrant, but also with their physical appearance and with their workplace. These authors also emphasized that migrant's health worsened after arriving in Spain, compared to their health in the country of origin.

Nationwide, Sevillano et al. [16] detected that perceived stress was the best predictor of physical and mental health. While this concept relates negatively to a sense of coherence and satisfaction with life, it relates positively to psychological distress and feelings of social exclusion. Some migrants under extreme migratory grief manifest somatizations, confusional and anxious-depressive symptoms, but also mourning due to separation, feelings of failure and loneliness, guilt, emotions associated with loss of status, the breaking of solidarity ties and extensive communities, fear of punishment, and hopelessness [20]. Furthermore, there is often an inadequate response of health care systems, and in a bureaucratic level, migrants have to often overcome challenging requirements concerning health treatment access and other basic means [21].

At the base of many of these health determinants, stress is defined as a person–environment, biopsychosocial interaction, wherein environmental events [stressors] are appraised first as unwanted and negative, and require some actions to cope with when adaptation fails [22]. The transactional stress model considers the existence of an interaction between the individual and the environment. The impact of the stressor firstly depends on the cognitive appraisal and the meaning given to it. Secondary appraisal assesses the abilities or resources available to cope with the event. If the individual interprets the stressor as negative or threatening, this evaluation predisposes the development of coping strategies. Folkman & Lazarus's [23] conveyed that coping refers to cognitive and behavioral responses individuals use to manage or tolerate stress. Additionally, the authors define two types of coping functions: the first is aimed at problem-solving, while the second is focused on reducing or managing emotional distress.

As commented above, perceived discrimination has been conceptualized as a stressor. However, each individual's response will be different depending on the previous perception of the stress situation [24]. Lahoz & Forns [25] found that people who perceived themselves as being discriminated against, tended to use more cognitive avoidance strategies to face the situation. Parallelly, other studies have found that identifying oneself with the underrepresented group can mitigate the negative consequences of racial prejudice and lead to a positive impact on well-being [26, 27].

Considering the complexity and diversity involved in the analysis of the migratory processes on mental health [28], migrant narratives were analyzed within a convenience sample aiming to describe their experiences of perceived discrimination and migration-related stress. Since significant differences have been found in association with different types of psychosocial vulnerability [29], gender and culture were considered when recruiting the sample for the study. It is hypothesized that perceived discrimination may increase the risk for mental health-related issues and the subsequent loss of emotional well-being. Participants who experience perceived discrimination may report difficulties associated with psychological distress, such as negative emotions, behavioral changes, discrimination-related stress, or mental health problems. Some individuals may refer to coping strategies to deal with the perceived discrimination.

In connection therewith, the goals of the present study are: 1) to identify the negative emotions, coping strategies, and behavioral changes related with perceived discrimination in newly-arrived migrants in Spain, and 2) to explore how perceived discrimination impacts their emotional well-being and mental health.

## 2. Materials and methods

### 2.1. Study design

The current study used data from the MigraSalud project [30]. The MigraSalud project of the Parc Sanitari Sant Joan de Déu [in collaboration with the Juan Ciudad Foundation] comprises

four independent studies that are part of a national scope financed by the Ministry of Labor, Migrations and Social Security in Spain. The project was born in 2018 to provide new scientific knowledge on post-migration well-being and health, with the ultimate purpose of contributing to society by improving healthcare practices. The present work is a qualitative study designed to understand the discrimination-related stress of Spanish migrants in the territory. The qualitative design was used due to the need to emphasize personal narratives and experiences to comprehend better the cultural reality. An in-depth interview instrument with 100 open-ended questions and 14 sections was created. The interview covers sociodemographic questions, employment conditions, the journey to Spain and the post-migration period, attachment, adverse life experiences, perceived discrimination and stress, social network, identity, intercultural mediation, public health care system in Spain, and COVID-19 pandemic.

## 2.2. Setting and sample

A snowball sampling strategy was used recruiting seeds from each social group to participate in the interviews and help identify other eligible newly-arrived migrants for recruitment. The recruitment channels were mainly non-profit organizations settled in three Spanish cities [Madrid, Valencia, and Barcelona]. Participants were also recruited from the MigraSalud project webpage [31], personal communications, and social media. The process of recruiting participants continued until thematic saturation was reached, which depended on including enough study participants (nearly 100) from the relevant autonomous communities (Madrid, Valencia, and Barcelona) in Spain and the newly arrived migrant vulnerable groups constrained by the languages available in our study.

The inclusion criteria were [1] being 18 years or older, [2] not being a Spanish citizen, [3] having lived in Spain for less than 5 years, [4] living, studying, or working in Barcelona, Madrid, or Valencia, and [5] speaking Spanish, English, Chinese, Urdu, French, or Arabic. In addition, the cities of Madrid, Barcelona, and Valencia were chosen since they are the most populated cities in Spain and can be compared between them [32].

## 2.3. Ethical statement

Ethical approval was provided by Parc Sanitari Sant Joan de Déu, Barcelona, Spain (PIC 41–20). Participants were thoroughly informed about the objectives and procedures of the study. Each respondent had to provide written informed consent before participation. All documents, including informed consent forms were available in English, Spanish, Chinese, Urdu, French, and Arabic.

## 2.4. Data collection

Data was collected through 102 individual audio-recorded in-depth interviews from February 2020 to November 2020. This sample size was deemed sufficient for the qualitative analysis and scale of this study, even so, reasons to stop recruitment included time and financial constraints. A guide for the semi-structured interview was developed specifically for this research and used by the facilitators. Questions vary from daily life discrimination experiences to health and emotional well-being concerns. Due to the COVID-19 pandemic, interviews were administered by phone or using online video-call platforms ensuring data encryption for security and privacy reasons. Five interviews were led in person, as some participants did not know how to use information technologies. During the interviews there were only the participant and the interviewer, to avoid the influence of the presence of non-participants. Consent forms and semi-structured interviews were available in English, Spanish, Chinese, Urdu, French, and Arabic because of the regional migration demographics [33]. Interviews lasted two hours

approximately (for more detailed information, see S2 Fig). Interviewers were native speakers and asked open-ended culturally appropriate questions. Before getting into the field, exhaustive training on how to conduct in-depth interviews was provided to the interviewers. Training on confidentiality, transcription, and translation processes was also ensured. Interviewers were also fluent in Spanish. Later, the interviewers transcribed the qualitative data in their native tongues, and translations in Spanish were handled. Usually, the same researcher who conducted the interview undertook the transcription and translation to the Spanish language itself; except for the Spanish language interviews, which were transcribed in some cases by a different researcher. Participants were interviewed in their native language and by an interviewer from a similar ethnic group. BMM, MCA-B, YH, REH-E, FV and H conducted the interviews, and they had no prior personal relationship with any of the participants [see S2 Table for more information].

Although 105 interviews were conducted, 3 interviews had to be excluded from the analysis due to recording problems that prevented its transcription [N = 2], and for not meeting the inclusion criteria [N = 1]; obtaining a final sample of 102 participants. As the participation was voluntary and the interview was performed in one session, the study did not have refusals or drop-outs.

## 2.5. Data analysis

Two researchers who did not participate in the interview or transcription process independently analyzed the participants' responses. Through conventional content analysis, they could gain information based on participants' unique experiences without imposing preconceived categories or theoretical perspectives. Independently, the two coders read and re-read the transcripts several times to achieve an understanding of the content of the interviews, while writing observations and highlighting text that appeared to be relevant to the aim of the study. As they read, they started to identify codes, based on patterns, similarities, differences, and relationships. Reflective remarks and memos were also used for the analysis. After independent open coding, the two coders met to discuss the preliminary codes upon reaching a consensus. Code definitions were used as the central criteria for assigning codes. Then, they kept coding by using these codes and adding new ones when they encountered data that did not fit into an existing code. During the coding process, the coders had weekly meetings to discuss new codes and group them into sub-categories and categories. In cases in which there were doubts or lack of context, coders also had meetings with interviewers (for more detailed information about the coding process, see S1 and S2 Figs). Agreement and in-depth discussions among researchers guaranteed intercoder reliability.

## 3. Results

### 3.1. Distribution of participants

Table 1 shows the characteristics of the overall study sample and by region of origin. Considering a non-binary gender approach, we recruited 48 (47.1%) self-identified women, 51 (50.0%) self-identified men, and 3 (2.9%) interviewees reported another gender identity. The age of the participants ranged from 18 to 57, with a mean age of 30.67 (SD = 9.81). Most of the participants had lived on average less than 3 years in Spain. Splitting participants by origin, 26 in-depth interviews were conducted with Chinese participants, 21 with Arabic participants from Morocco, 6 with interviewees from other African countries (i.e., Argelia, Cameroon, Mali, Nigeria, Senegal, and Sudan), 40 participants were from Latin America (i.e., Argentina, Brazil, Chile, Colombia, Cuba, Dominican Republic, Honduras, Mexico, Peru, and Venezuela), and 10 from Pakistan and India. Most participants (94 of 102) reported having experienced some type of discrimination since their arrival in the host country. Those who did not report having

**Table 1. Characteristics of the overall study sample and description of the participants by origin.**

| Characteristic | Total sample [N = 102] | Latin America [n = 40] | Africa [n = 26] | China [n = 26] | Pakistan & India [n = 10] |
|---|---|---|---|---|---|
| **Gender**, n[%] | | | | | |
| Male | 51 [50.0] | 15 [40.0] | 23 [88.5] | 6 [23.1] | 6 [60.0] |
| Female | 48 [47.1] | 23 [57.5] | 2 [7.7] | 19 [73.1] | 4 [40.0] |
| Other | 3 [2.9] | 1 [2.5] | 1 [3.9] | 1 [3.9] | 0 [0.0] |
| **Age**, mean[SD] | 30.67 [9.81] | 32.58 [9.08] | 24.16 [9.15] | 32 [8.68] | 35.9 [10.47] |
| **Education**, n[%] | | | | | |
| Primary | 11 [10.9] | 2 [5.0] | 6 [24.0] | 0 [0.0] | 3 [30.0] |
| Lower-secondary | 24 [23.8] | 4 [10.0] | 14 [56.0] | 4 [15.4] | 2 [20.0] |
| Higher-secondary | 21 [20.8] | 13 [32.5] | 3 [12.0] | 3 [11.5] | 2 [20.0] |
| Tertiary | 45 [44.6] | 21 [52.5] | 2 [8.0] | 19 [73.1] | 3 [30.0] |
| **Partner**, n[%] | | | | | |
| No | 68 [66.7] | 26 [65.0] | 22 [84.6] | 16 [61.5] | 4 [40.0] |
| Yes | 34 [33.3] | 14 [35.0] | 4 [15.4] | 10 [38.5] | 6 [60.0] |
| **City**, n[%] | | | | | |
| Barcelona | 62 [60.8] | 8 [20.0] | 20 [76.9] | 25 [96.2] | 9 [90.0] |
| Madrid | 20 [19.6] | 16 [40.0] | 3 [11.5] | 1 [3.9] | 1 [10.0] |
| Valencia | 20 [19.6] | 16 [40.0] | 3 [11.5] | 0 [0.0] | 0 [0.0] |
| **Time living in Spain [years]**, mean[SD] | 2.84 [1.60] | 2.40 [1.73] | 2.50 [1.20] | 3.31 [1.44] | 4.27 [1.43] |

Frequencies and proportions [in percentages] are displayed for categorical variables, and means with standard deviation [SD] for continuous variables.

been discriminated against when asked directly, later in their narrative, it was revealed that they had experienced some form of discrimination. However, it had not been perceived as such. This could be attributed to a lack of knowledge of the concept, not being aware of the phenomenon, or due to stigmatization, as some participants reported that for them, it meant having low self-esteem and a debility trait. An example of this experience was described by Participant 18 [P18]:

> P18: *I've never experienced it (discrimination), but similar things have happened to me, for example there's people in the street with their phone and they are scared or walk away from the Moroccan [referring to himself], or you make a question to someone, and he/she doesn't answer. . . .*
>
> [18 years, from Morocco]

### 3.2. Findings

Five emerging categories were found post-hoc related to the aim of the present study: [1] "Negative emotions due to the perceived discrimination", [2] "Change in behaviors due to the perceived discrimination", [3] "Identified Coping Strategies to face Discrimination-Related Stress", [4] "Discrimination-related Stress", and [5] "Mental Health Problems due to Discrimination-related Stress". A total of 48 codes were agreed by the two researchers included in the present study. The 11 sub-categories that emerged from the coding process facilitated the categorization into the 5 main categories (see S1 Table). Most of the participants openly communicated to have experienced negative emotions due to the perceived discrimination, and some of

them reported a decrease in their functional well-being. In general, it was reported that most of the participants changed their behaviors after perceiving discrimination and many of them were also able to identify coping strategies to deal with the discrimination phenomenon. Not all the participants linked any health problem or emotional distress to their perceived discrimination. However, when the interviewers dug into some discrimination narratives it was possible to distinguish new descriptions of discrimination-related stress and mental health problems.

**3.2.1. Negative emotions related to the perceived discrimination.** When participants were asked about their thoughts, feelings, and emotions just after being discriminated against, the majority of them reported negative emotions. Overall, when a person is subject to discrimination, their experience becomes unique and unrepeatable. Even though the range of emotions varies from one person to another, it has been possible to identify key feelings and emotions that are more prevalent in the narratives of the people interviewed.

**'Disgust'**, **'sadness'**–which contains disappointment and feeling bad–, and **'fear'**–which includes worry, anxiety, and fear to be discriminated against–were described when the participants felt that although they did not do anything wrong, they could not avoid being discriminated. **'Loneliness'**–including feeling lonely, marginalized, misunderstood, rejected, and abandoned–, **'humiliation'**–which embraces also feeling offended or denigrated–, **'hypervigilance'**–containing the sense of being judged or watched by others before acting–and **'mixed'** emotions [i.e., when one or more feelings, emotions, or moods occur together] were also stated by some participants, like participant 1 [P1]:

> P1: *When this happens to me [being discriminated] I feel bad all day. It also makes me worry about something similar happening to me again.*
>
> [26 years, from China]

**'Shame'** and **'guilt'** were some of the emotions conveyed mainly by women from the Chinese community, usually after being excluded or ridiculed by a group of people from their work or study place. The codes **'sense of injustice'** and **'rage'**–which includes anger, revenge, or impotence–was repeatedly expressed by young men from African countries, and was related to discriminations that mainly happened in public transportation:

> P2: *Once, some friends sneaked into the subway because they didn't have transportation tickets, so the subway security guards cursed us and asked for our documents, and they started saying 'these MENAS [Spanish acronym for unaccompanied minors used derogatorily], they are criminals', and that is not true, and neither is the word MENAS fair. . . and then I must keep quiet, because if I speak without witnesses for the facts, I won't get anything. . .*
>
> [19 years, from Morocco]

Lastly, almost all interviewees described that they changed the way they felt about themselves. Some of the most repeated feelings were to **'feel undervalued or inferior'** [i.e., feeling worthless or that others do not value you as they should] [quote by P3] and to **'feel vulnerable'**, which was mainly attributed to being undocumented and to the lack of institutional or social supports [quote by P4].

> P3: *. . .when we get on the public bus, people move away from us, it's obvious. [. . .] I feel inferior, I feel like a shit, they treat me like I'm disgusting.*
>
> [35 years, from Cameroon]

P4: *I feel very bad because we don't have any relatives who can defend us, nor acquaintances. We don't have anyone who can help us to denounce it or anything. I am feeling bad. And that's all about it.*

[18 years, from Morocco]

**3.2.2. Change in behaviors due to the perceived discrimination.**   Perceived discrimination not only causes immediate negative emotions but also makes participants acquire changes in their behaviors to avoid being discriminated against in the future.

Some changes in behavior mainly seen in individuals from Pakistan, China, and Morocco focused on being more accepted by the local community [quotes by P5 & P6] by **'westernization or cultural assimilation'** (i.e., changing their lifestyle and their dress code to look like a local and attract less attention) and by **'creating a good image'** of oneself and its culture.

P5: *There are people [referring to migrants] who steal and others who don't, but people [referring to locals] see all of us the same. . . especially in the subway. I tell to my colleagues that if someone stares at you badly, you just ignore it to avoid problems. I also advise them to change the way they dress and their hairstyle, because unfortunately, people glance at you for your physical appearance instead of looking at you for your behavior or your sufferings in life.*

[19 years, from Morocco]

P6: *They have no respect for Islam. People don't speak well about my religion, so I avoid conflict situations, for example I don't have a beard because it is not well seen.*

[47 years, Pakistan]

Changes in behavior to avoid discriminatory situations such as **'not going out'**, **'avoid going alone'** [quote by P7], **'hypervigilance'** [i.e., change routes or places where they used to go and be very careful and more concerned over what they say or do], and **'stop participating in politics'** were mentioned. Additionally, a participant from Senegal also noticed that he had become more **'self-sufficient'**, showing less caring or solidary with others.

P7: *The lesson learned from this experience is to avoid sitting alone in the subway. I have to be in a crowd of people. If I am alone, I am more likely to be offended without company or support around me.*

[22 years, from China]

Finally, few participants adopted extreme behavioral changes. On one hand, **'positive adaptation'** was assimilated by some individuals as a healthy functioning posture [quote by P8], while a couple of participants acquired a negative adaptation by **'becoming the oppressor'** and repeating the same patterns of their bullies.

P8: *Thanks to them [people who discriminated her] I can better manage the stress and pressure of discrimination. The other day I defended another person against a thief on the street. If it were earlier, I would not have had the courage.*

[26 years, from China]

**3.2.3. Identified coping strategies to face discrimination-related stress.**   Once the participant identified a discriminatory experience, the interviewers inquired about the coping

strategies the person put in place to deal with it. Generally, participants have their singular manner to deal with perceived discrimination afterwards, but parallel strategies within members of the same nationality background were distinguished.

Internalized coping strategies were defined by the research team as those strategies that did not require interaction with others. As internalized coping strategies, participants conveyed **'ignoring'** or **'not responding'** to the act or acts of discrimination. Reasons described by the participants were, for instance, fear of getting legal documents revoked or not obtaining them for possible accusations of uncivil behavior. Moroccan unaccompanied minors like P9, felt extremely vulnerable and mainly used these coping strategies:

P9: *Because if you don't know how to speak the language, and you are undocumented, it's better to ignore and that's it. . .*

[19 years, from Morocco]

Chinese individuals in the academic context repeatedly reported **'isolation'** as an adopted coping strategy–which includes detaching oneself from the bully or bullies:

P1: *I isolate myself observing everything. For example, when I was abroad on my Erasmus time, I just left my place and find another one where I felt more accepted.*

[26 years, from China].

A small number of individuals from a variety of nationalities mentioned **'self-medication'** to deal with daily life discrimination:

P10: *I am not taking alcohol anymore, but I am taking sleeping pills again. I take them every night before sleeping time. Then, I get up as if I were a ghost; you know my body feels dead. I do some exercise to sleep, but that does not help me a lot.*

[24 years, from Sudan]

Finally, mainly Chinese participants reported **'intellectual activities, leisure and sport'**– that includes a range of internalizing strategies as writing an introspective diary, reading, listening to music, exercising, and cognitive training.

Externalized coping strategies were operationalized as social behaviors that implied others. Externalized coping strategies included **'to talk with the oppressor'** (i.e., to search for a constructive dialogue with the bully and request to solve common problems), **'to insult the oppressor'**–that mainly happened in the street with an unknown attacking person–, **'to denounce'** [e.g., using social media platforms, recording the discrimination situation in their phones to obtain public attention, calling the police, etc.], **'physical fight'** or **'revenge'** by attacking the bully, **'to seek for comfort'**–in this case trusting a meaningful person [quote by P11]–, and **'to increase solidarity with others'**.

P11: *I seek comfort with my boyfriend. My boyfriend has many experiences managing social situations and interactions, and with him, I can seek comfort*

[31 years, from China]

Mixed coping strategies were identified used individually or in a social context. For example, **'to cry'**, as some individuals cried in isolation and others–mainly from Latin-American

communities–preferred to share their crying with somebody else, and **'to use humor'**, as some of the participants used this resource as an internal managing thought, while others external-ized the use of humor by engaging with another person.

**3.2.4. Discrimination-related stress.**    Some of the interviewees reported that being dis-criminated against increased their emotional distress response:

> P12: *I am feeling more stressed out and nervous. . . when I think about it, I just want to ignore everything.*
>
> [18 years, from Morocco]

> P13: *All these issues are generating a huge emotional and mental distress to me. Very huge [emphasizing the word huge]. To begin with, it was very difficult the relationship with my tutor. In fact, he is the most important person which whom I have to interact here abroad. Our interpersonal relationship is by far the most difficult for me. For this reason, it generates me a lot of emotional and mental distress, I have to think about each word I say, everything I do, and I have to behave carefully. Recurrently, I have to avoid his comments, and I have to be diplomatic with him constantly.*
>
> [32 years, from China]

**3.2.5. Mental health problems due to discrimination-related stress.**    Participants dis-closing discrimination-related stress described mostly mental health-related problems.

**'Problems related to substance abuse and addictive behaviors'** were conveyed by some of the interviewees–for example, **'alcohol and marihuana abuse'** [quote by P14], **'misuse of anx-iety medications'**, and **'compulsive video gaming'**.

> P14: *I get recurrent nightmares because of my stress. The way I deal with it is drinking, and complaining with friends.*
>
> [35 years, from China]

**'Problems related to mood'** englobes codes as **'sadness'**, **'depression'**, and **'suicidal thoughts'**, as explained by P15:

> P15: *I feel discriminated, and I feel very bad. I feel like a dog. My neighbor's name is Juan [typ-ical Spanish name] but my name is [an Arabic name]. Sometimes I think about suicide, because I feel very humiliated.*
>
> [46 years, from Morocco]

**'Problems related to anxiety'** were also described. Anxiety was operationalized including also overthinking and over worry. Some of the participants used the word **'anxiety'**, but others, especially from Morocco where the term is not culturally present, used expressions as **'nervousness'** or **'having nerves'**. Behavioral symptoms of anxiety as **'nail biting'** were pointed out. **'Eating compulsively'** and **'sleeping problems'** narratives were also outlined. In the fol-lowing case, a man was talking about the systemic discrimination experienced and how it affected his health:

> P16: *Even if you are an undocumented migrant; you deserve a health care access option. Like everyone else, right? If I have to pay for health care services it is not a problem, but right now,*

*it costs a lot [. . .] so, what can I do? I am afraid. I can't get my checkups done because I can't afford this option. . .*

[43 years, from Argentina]

Another participant disclosed **'psychological enuresis'** as an anxiety problem:

P10: *I recurrently pee in my bed during nighttime, you know? Sometimes I am afraid of everyone, I can't take the bus, I can't walk around the streets, I feel everyone is against me, you know? Everyone is against me . . .*

[24 years, from Sudan]

Lastly, **'not classified problems'** as for example **'headaches'** were conveyed:

P17: *Yes, this situation is really creating stress on me, and. . .well, I begun having headaches.*

[34 years, from Argentina]

As a general principle, physical health issues were not directly linked with the stress of being discriminated against. Contrarily, a connection between mental health narratives and perceived discrimination existed. Specifically, physical health problems were associated with immigration issues as harsh labor conditions [e.g., having body or back pain]. The only remarked situation where it was described as a physical condition related to the perceived discrimination was when talking about headaches by a Latin American individual.

## 4. Discussion

### 4.1. Main findings

The data indicate that perceived discrimination negatively impacts the well-being and mental health of newly-arrived migrants in Spain. Different negative expressions are narrated in a broad continuum, ranging from negative emotions, psychological distress, and more severe mental health-related concerns. In this qualitative study, data acknowledged that newly-arrived migrants in Spain have internal, external, and mix coping strategies to deal with perceived discrimination. As a main finding, changes in their form of behaving may occur once they perceive unfair treatment.

### 4.2. Interpretation

The interviews revealed several negative emotions. Similar findings were reported by Crocker [34], who found emotional suffering linked to stress, loneliness, fear, depression, and trauma in migrants, which have been documented to contribute to mental health risk [35].

In this same line, 'shame' or 'guilt', 'sense of injustice', or need for 'vengeance' reported by the sample of the study, are secondary emotions that imply feeling something about the primary feelings of 'rage' or 'anger' expressed as related to discrimination. As stated by Braniecka et al. [36] these outputs are useful in terms of functional interpretation and emotional adaptation as they may facilitate adaptive coping by promoting the motivational and informative functions of emotions. Fostering solution-oriented actions instead of avoidance, facilitating insight access, better narrative organization, and providing a resilient attitude towards stress and difficult experiences may also ease adaptive coping [36].

The results obtained in terms of coping strategies to face perceived discrimination and the related stress are not based on any pre-established scale, which has allowed to obtain responses

without any preconceived ideas. Most coping strategies mentioned by the interviewees largely match with categories established in previous studies, like the eight dimensions described by Folkman et al. [37] [i.e., confrontive coping, distancing, self-controlling, seeking social support, escape–avoidance, planful problem-solving, positive reappraisal, and accepting responsibility]. The dimension "accepting responsibility" was the only one not represented in the study.

Despite previous studies have analyzed the coping strategies used by a migrant community in a concrete country [6, 27], few investigations have focused on the different forms of coping between distinct nationalities that coexist in the same context. Although this research has been able to discern some diversity in the coping approaches of the studied communities, further investigation is needed to delve into the nature of these differences.

The manifestations of 'westernization or cultural assimilation' or the efforts described to 'create a good image' found in the present analysis, have also been identified by Bhugra & Becker [38], who studied how the individual's cultural identity may be lost during the assimilation process within the host society that follows acculturation [39]. Acculturation frequently results in stress, self-esteem problems, and mental health damage [40]. Changes in behaviors to avoid being discriminated against were part of the narrative repeated throughout the interviews. Individuals in the present study described feeling 'vulnerable, worthless, inferior, or undervalued.' These concepts have been explained by Leary & Springer [41] as the result of interactions that involve relational devaluation that often causes hurt feelings. This could also illustrate the results in the feelings of lack of social support perceived by the interviewed participants and the reaction of changing to a 'self-sufficient' attitude as a defense mechanism against discrimination contexts and events.

The 'emotional distress response' expressed by the participants as a reaction to discrimination has been previously acknowledged by Wallace et al. [42]. The author reported migrants feeling unsafe and vigilant. Migrants also showed anticipatory stress of a possible future racist encounter. Anticipatory stress increased the probability of avoiding spaces suggesting that past exposure to racial discrimination or awareness of racial discrimination experienced by others can continue to affect individual's mental health after arriving in the host country.

Concerning the mental health outcomes raised by the participant's answers, 'problems related to substance abuse and addictive behaviors' have been captured by other authors. For instance, Borges et al. [43] found patterns of substance use disorders when facing loneliness, social isolation, stress, and discrimination linked to broader social changes associated with transnational migration, naming direct exposure to substance use opportunities, the transfer of social norms of substance use, and the economic means as factors involved. Horyniak et al. [44] also found regional and global differences in patterns of 'substance use', which may be influenced by local context factors such as availability of substances and social norms.

The mood symptoms expressed by the interviewed sample in this study agree with the literature evidence of the last decade. Fortuna et al. [45] and Wolf et al. [46] exposed experiences of 'depression, trauma exposure, pessimism, sense of failure, guilt feelings, punishment feelings, and suicidal thoughts'. The present results also support previous findings related to 'anxiety problems'. Szaflarski et al. [47] explain them as related to stress and the preference for socializing outside one's racial-ethnic group, while in another study, Sapmaz et al. [48] and Mares [49] show varying rates of feeding and sleeping problems, nail-biting, enuresis, and other regressive symptoms among children, as the present investigation does with adults.

In sum, perceived discrimination may constitute a risk factor for mental health deterioration and psychological distress. Most of the experiences described by the target sample in the present study match the above findings where the frequency of perceived discrimination events negatively relates to well-being levels in the migrant population [3] and different coping strategies are displayed to face these experiences.

## 4.3. Strengths and limitations

**4.3.1. Convenience sampling.**    This study has several strengths and limitations. Convenience sampling was chosen as the recruitment method as it is cost and time-efficient, and simple to implement in one-year funded projects [50]. Because the thematic was relatively new in Spain, and scarce literature on the topic was available in the Spanish context, the team wanted to begin from scratch and ask newly-arrived migrants directly about discrimination and mental health. Nevertheless, this sampling method lacks generalization, and the results cannot be transferred into the general Spanish population or between different cultural communities of newly-arrived migrants in Spain [50]. Although the present study focused in three cities that are similar in terms of immigration, they also present some local particularities [such as the presence of co-official languages in Barcelona and Valencia] that may have an impact on integration process due to language barriers.

**4.3.2. Recruiting.**    Considering that convenience sampling is often not reflective of the target population [50], it is essential to mention the limitations in this regard. In general, older adults are underrepresented in this research. This fact may be because of the recruitment channels used, tied to particular age-group individuals. Moreover, Idescat [51] indicates that registered migrants in Catalonia from the countries interviewed average age is 30 to 44 years old. The tax rate of migrants of more than 60 years old in Catalonia is low–less than 10%–. These numbers may be connected with the idea that Spain is a relatively new host country. Idescat's [51] data is salient as most of the interviews were conducted in the Catalan territory, except for the interviewees from Latino America [N = 38] that were mainly widespread in Valencia [N = 16] and Madrid [N = 13].

With an intersectional perspective, individuals recruited in the present study were from different nationalities, races, ethnicities, classes, religions, cultures, gender identities, and sexual orientations. In the Latino-American sample, diversity in gender and sexual orientation is well represented [N = 6]. The team had difficulties incorporating disability in the sample [N = 1].

The underrepresentation in terms of gender in some communities hinders the identification of possible gender differences. Specifically, looking into the sample from newly-arrived Chinese migrants, there was a predominance of women [N = 19] with high-level college degrees with a stable financial situation. As a great strength, interviews were conducted by a culturally aware native Chinese speaker. The sample from Morocco has its constraints too. It was mainly composed of male unaccompanied minors transitioning to adult life from foster care placements in Barcelona [N = 15]. In future studies, more self-identified women should be interviewed to explore gender issues regarding stress and discrimination among this community. A main strength was that the interviewer is a social worker from Morocco who has been working with unaccompanied minors for several years and carried out the interviews in the Moroccan dialect of Arabic, known as Darija. The same researcher interviewed the other African countries' participants, as he speaks Arabic and French, and shared cultural and religious background with some of the interviewees. The final sample from African countries [excluding Morocco] was tiny [N = 6], as well as the selection from Pakistan and India [N = 10], being a major barrier for the study of these specific communities. A new manner to reach individuals from these countries must be explored in future studies, as this research was conducted during the COVID-19 pandemic, and challenges with the uses of technology emerged.

**4.3.3. Data collection and analysis.**    Our results should be interpreted considering some data collection and analysis limitations. First, the one-to-one limitation linked to the interviews, as the obtained data depends mainly on the interviewer's sensitivity and persistence, and interpersonal interaction. Second, our study was tied to perform most of the interviews

online, instead of face-to-face, due to the COVID-19 pandemic social distancing measures. Interviews through online platforms might have excluded people without internet access, a computer, a mobile phone, or unfamiliar with telematic tools. Finally, it is important to highlight that mental health problems due to discrimination-related stress disclosed by the participants were self-reported. Self-reported data is linked to social desirability bias, recall bias, and can be limited by the introspective ability of the participants and their personal bias in relation with the topic. Nonetheless, our study provides an important addition to the literature regarding perceived discrimination in Spain and its consequences on health. Considering the lack of previous research on the topic, the used qualitative methodology allows capturing the complexity of people's behavior and their health issues by obtaining direct information from study participants without imposing preconceived categories or theoretical perspectives before the analysis [52].

## 4.4. Implications, macro level interventions and areas of future research

Qualitative research can find suitable answers for health and social care policy-makers and professionals. The study presented and the data available until today show that discrimination is becoming a significant public health problem [4, 53]. The present research reaffirms that discrimination is connected to negative emotions, distress, and other mental health challenges. These have some implications as the need to invest in the Spain migrant population's mental well-being and the promotion of mental health care access and resources within the community. Accessibility to mental and social services will help to detect discrimination cases in advance. Mental health care access is necessary, but identifying discrimination needs to encompass a multidisciplinary and sensitive cultural team of professionals. An antiracist curriculum and culture within the agencies are essential to cover all the aspects of the practice with newly-arrived migrants in a culturally responsive form. Psychological support is critical for preventing mental health challenges, but legal aid needs to exist in parallel to detect and manage discrimination in a social justice-oriented form. A human rights-based perspective in the clinical and social practice that supports the existing Declaration of Human Rights, and the present European Human Rights treaties will improve the current public health issue. The newly-arrived migrants, in conjunction with local people, must empower themselves to identify different types of discrimination to create common synergies of respect, tolerance, and advocacy towards social change. This work must be mutually reciprocal.

To create and promote this anti-discrimination programs and policies, it is relevant to comprehend that each interviewed community and culture perceive discrimination differently. This is because the stereotypes linked to migrants varied depending on their nationality, skin color, gender, sexual orientation, social class, weight, and religion. Accordingly, an intersectional analysis is needed to propose new policy improvements within the Spanish social system. Programs and policies need to adapt specifically to the different communities, be antiracist and inclusive, and social needs must be considered integrating the narratives and life stories of newly-arrived migrants. Individuals from the same community need to be actively involved in creating and implementing programs and policies, with peer support or coordinating efforts with thematic experts. Alongside, policies and awareness-raising campaigns must cover different fields of action and disciplines, as schools, universities, public institutions, or state security forces.

Future research must approach the pandemic effects of the COVID-19 and the convergence of discrimination, stress, and mental health in newly-arrived migrants. Today, and more than ever, this pandemic is showing its effects as travelling restrictions arise and border closing solidify. Considering the Real Decreto declaration 463/2020 released by the Spanish Ministry

of Inclusion, Social Security and Migrations [54], subsequent studies must not forget how bureaucracy and procedures regarding legal status have been frozen in Spain during the pandemic.

## Supporting information

**S1 Table. Description of codes and categories used in the present study.**
(DOCX)

**S2 Table. Sociodemographic characteristics of interviewers.** Note: S = Spanish; E = English; U = Urdu; A = Arab; CH = Chinese.
(DOCX)

**S1 Fig. Illustrative description of the coding process.**
(TIF)

**S2 Fig. Example of the coding process.**
(TIF)

## Acknowledgments

The authors would like to express special gratitude to all the participants for their generous contribution, which made this study possible.

## Author Contributions

**Conceptualization:** Aina Gabarrell-Pascuet, Amanda Lloret-Pineda, Blanca Mellor-Marsa, Maria del Carmen Alos-Belenguer, Yuelu He, Rachid El Hafi-Elmokhtari, Paula Cristóbal-Narvaez.

**Formal analysis:** Aina Gabarrell-Pascuet, Amanda Lloret-Pineda.

**Funding acquisition:** Josep Maria Haro, Paula Cristóbal-Narvaez.

**Investigation:** Aina Gabarrell-Pascuet, Amanda Lloret-Pineda, Blanca Mellor-Marsa, Maria del Carmen Alos-Belenguer, Yuelu He, Rachid El Hafi-Elmokhtari, Felipe Villalobos, Ivet Bayes-Marin, Josep Maria Haro, Paula Cristóbal-Narvaez.

**Supervision:** Josep Maria Haro, Paula Cristóbal-Narvaez.

**Writing – original draft:** Aina Gabarrell-Pascuet, Amanda Lloret-Pineda, Marta Franch-Roca, Blanca Mellor-Marsa, Maria del Carmen Alos-Belenguer, Paula Cristóbal-Narvaez.

**Writing – review & editing:** Aina Gabarrell-Pascuet, Amanda Lloret-Pineda, Marta Franch-Roca, Blanca Mellor-Marsa, Maria del Carmen Alos-Belenguer, Yuelu He, Rachid El Hafi-Elmokhtari, Felipe Villalobos, Ivet Bayes-Marin, Lola Aparicio Pareja, Oscar Álvarez Bobo, Mercedes Espinal Cabezas, Yolanda Osorio, Josep Maria Haro, Paula Cristóbal-Narvaez.

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
