## [Decision Letter · Decision Letter 0]

1 Aug 2023

PONE-D-23-16852Impact of Perceived Discrimination and Coping Strategies on Well-being and Mental Health in Newly-Arrived Migrants in SpainPLOS ONE

Dear Dr. Cristóbal Narváez,

Thank you for submitting your manuscript to PLOS ONE. After careful consideration, we feel that it has merit but does not fully meet PLOS ONE’s publication criteria as it currently stands. Therefore, we invite you to submit a revised version of the manuscript that addresses the points raised during the review process. Please submit your revised manuscript by Sep 15 2023 11:59PM. If you will need more time than this to complete your revisions, please reply to this message or contact the journal office at plosone@plos.org. Please include the following items when submitting your revised manuscript:A rebuttal letter that responds to each point raised by the academic editor and reviewer(s). You should upload this letter as a separate file labeled 'Response to Reviewers'.A marked-up copy of your manuscript that highlights changes made to the original version. You should upload this as a separate file labeled 'Revised Manuscript with Track Changes'.An unmarked version of your revised paper without tracked changes. You should upload this as a separate file labeled 'Manuscript'.If applicable, we recommend that you deposit your laboratory protocols in protocols.io to enhance the reproducibility of your results. Protocols.io assigns your protocol its own identifier (DOI) so that it can be cited independently in the future. For instructions see: https://journals.plos.org/plosone/s/submission-guidelines#loc-laboratory-protocols. Additionally, PLOS ONE offers an option for publishing peer-reviewed Lab Protocol articles, which describe protocols hosted on protocols.io. Read more information on sharing protocols at https://plos.org/protocols?utm_medium=editorial-email&utm_source=authorletters&utm_campaign=protocols.

We look forward to receiving your revised manuscript.

Kind regards,

AKM Alamgir, PhD

Academic Editor

PLOS ONE

“ALP, BMM, MCAB, YH, RHE, FV, and IBM work was endorsed by the project (II IN 200803 EN 162 FA 01), financed by the Spanish Ministry of Labor, Migrations and Social Security with the support of the European Commission (https://www.inclusion.gob.es/web/migraciones/fami). AGP work was financed by the Secretariat of Universities and Research of the Generalitat de Catalunya and the European Social Fund (2021 FI_B00839) (https://agaur.gencat.cat/en/beques-i-ajuts/Research-grants/ajuts-predoctorals/index.html). PCN's work was supported by Sara Borell (CD20/00035) and financed by the Instituto de Salud Carlos III (https://www.isciii.es/QueHacemos/Financiacion/solicitudes/Paginas/default.aspx). The funders had no role in study design, data collection and analysis, decision to publish, or preparation of the manuscript.”

“Amanda Lloret-Pineda’s, Blanca Mellor-Marsa’s, Maria del Carmen Alos-Belenguer’s, Yuelu He’s, Rachid El Hafi-Elmokhtari’s, Felipe Villalobo’s, and Ivet Bayes-Marin work was endorsed by the project (II IN 200803 EN 162 FA 01), financed by the Spanish Ministry of Labor, Migrations and Social Security with the support of the European Commission. Aina Gabarrell-Pascuet work was financed by the Secretariat of Universities and Research of the Generalitat de Catalunya and the European Social Fund. Paula Cristóbal-Narváez's work was supported by Sara Borell (CD20/00035) and financed by the Instituto de Salud Carlos III.”

“ALP, BMM, MCAB, YH, RHE, FV, and IBM work was endorsed by the project (II IN 200803 EN 162 FA 01), financed by the Spanish Ministry of Labor, Migrations and Social Security with the support of the European Commission (https://www.inclusion.gob.es/web/migraciones/fami). AGP work was financed by the Secretariat of Universities and Research of the Generalitat de Catalunya and the European Social Fund (2021 FI_B00839) (https://agaur.gencat.cat/en/beques-i-ajuts/Research-grants/ajuts-predoctorals/index.html). PCN's work was supported by Sara Borell (CD20/00035) and financed by the Instituto de Salud Carlos III (https://www.isciii.es/QueHacemos/Financiacion/solicitudes/Paginas/default.aspx). The funders had no role in study design, data collection and analysis, decision to publish, or preparation of the manuscript.”

Reviewers' comments:

Reviewer's Responses to Questions

**Comments to the Author**

1. Is the manuscript technically sound, and do the data support the conclusions?

Reviewer #1: Yes

Reviewer #2: Yes

2. Has the statistical analysis been performed appropriately and rigorously? 

Reviewer #1: Yes

Reviewer #2: N/A

3. Have the authors made all data underlying the findings in their manuscript fully available?

Reviewer #1: Yes

Reviewer #2: Yes

4. Is the manuscript presented in an intelligible fashion and written in standard English?

Reviewer #1: Yes

Reviewer #2: Yes

5. Review Comments to the Author

Reviewer #1: The article necessitates some grammatical revisions, particularly concerning the prohibition of commencing sentences with numerals. Please review the article for a few more minor grammatical errors. Moreover, the term "health problems" should be replaced with "mental health problems" to accurately reflect the focus of the article, which pertains to mental health.

The authors have gathered data on variables such as age, gender, education, partnership status, city of residence, and average duration of residency in Spain. Subsequently, did they examine whether any significant associations exist between these variables and their influence on the mental health of the participants?

Reviewer #2: The authors of this manuscript discuss the effects of perceived discrimination on the emotional well-being and mental health of new immigrants and the response mechanisms they used to deal with the perceived discrimination. This study specifically focuses on the context of new immigrants in Spain.

The PLOS ONE outlines that qualitative studies should be reported in accordance with the COREQ or SRQR Checklists. The COREQ checklist was used for this review. Unless otherwise noted, the authors satisfied the criteria of the checklist.

Specific Comments:

•Generally, the manuscript is well written and the study design is explained in detail. The relation of perceived discrimination as a stressor is appropriate given other literature on the topic. The introduction is clear in establishing the link of perceived discrimination and stress to physical and mental health implications in the migrant population.

•Section 2.4 Data Collection (starting at line 173) should include clear information about the interviewers’ personal characteristics and relationship with participants (COREQ #1-4, 6-8).

•Line 194-195, the authors indicate interviews which were excluded from the analysis. However, if there was the occurrence of refusals or drop outs, the authors did not indicate as such (COREQ #13). Nor, was the presence of non-participants identified (COREQ #15).

•The authors appropriately address the limitations of the study in the strengths and limitations section of the manuscript.

•Line 602-603, the last sentence is not necessary “Besides, it would be interesting to transfer the study findings to other stigmatized and oppressed groups in future studies.”

Major Comments:

•To better understand the diversity of the study population, the authors should comment on the reason of immigration (e.g. forced migration, economic, etc.). In a related capacity, the manuscript does not comment how this factor may impact the findings of the study. Were there any differential experiences based on this factor? Some explanation is required here.

•In section 3.1 Distribution of Participants (line 221-222), the authors write “94 of 102 participants reported having experienced some type of discrimination since their arrival to the host country”. What were the experiences of the other 8? Was it revealed during the interview they in fact experienced some form of discrimination but did not perceive it as such? How were those interviews considered in the analysis and how might this influence the findings? Some explanation is required here.

6. PLOS authors have the option to publish the peer review history of their article (what does this mean?). If published, this will include your full peer review and any attached files.

Reviewer #1: **Yes: **Kawalpreet Kaur

Reviewer #2: No

---

## [Author Response · Author response to Decision Letter 0]

28 Sep 2023

Thank you for the opportunity to improve the manuscript PONE-D-23-16852 entitled “Impact of Perceived Discrimination and Coping Strategies on Well-being and Mental Health in Newly-Arrived Migrants in Spain”. We would like to express our sincere gratitude for the suggested corrections. We have addressed all issues raised by the reviewers and revised the Journal’s requirements. 

Author's Reply to Reviewer 1

1) The article necessitates some grammatical revisions, particularly concerning the prohibition of commencing sentences with numerals. Please review the article for a few more minor grammatical errors. 

Reply: Thank you for your comment, we have revised the article for grammatical errors, and we have specifically corrected the sentences commencing with numerals:

Line 237: “Most participants (94 of 102) reported having experienced some type of discrimination since their arrival to the host country.”

Line 232: “Splitting participants by origin (Figure 1), 26 in-depth interviews were conducted with Chinese participants, 21 with Arabic participants from Morocco, 6 with interviewees from other African countries (i.e., Argelia, Cameroon, Mali, Nigeria, Senegal, and Sudan), 40 participants were from Latin America (i.e., Argentina, Brazil, Chile, Colombia, Cuba, Dominican Republic, Honduras, Mexico, Peru, and Venezuela), and 10 from Pakistan and India.”

Line 226: “Considering a non-binary gender approach, we recruited 48 (47.1%) self-identified women, 51 (50.0%) self-identified men, and 3 (2.9%) interviewees reported another gender identity.”

2) Moreover, the term "health problems" should be replaced with "mental health problems" to accurately reflect the focus of the article, which pertains to mental health.

Reply: Thank you for noticing this; we have corrected the terminology. 

3) The authors have gathered data on variables such as age, gender, education, partnership status, city of residence, and average duration of residency in Spain. Subsequently, did they examine whether any significant associations exist between these variables and their influence on the mental health of the participants?

Reply: Thank you for your comment. The codes identified arose from the participants’ very diverse narratives. Therefore, each unique code was not represented by a sufficiently significant number of people to associate it with specific variables. Furthermore, as we reported in the Limitations section, since it was a convenience sample, the recruitment of participants focused on vulnerable populations who generally share similar sociodemographic characteristics. Most groups correspond to the same community; for example, the unaccompanied minors are mainly from Morocco, the higher education and socioeconomic level sample is mainly from China, and domestic workers are from Latin America. This means that the only associations we could find were qualitative and mainly linked to the country of origin. The fact that, for example, the sample from Morocco was mainly comprised of young men, could lead us to say that codes associated with this group are associated with young people and the male gender, but this would be a biased statement. Therefore, we believe that due to the type of analysis and sample, we cannot determine these associations. We believe that a future quantitative study could better explore these associations, which would be very interesting to improve understanding of this phenomenon. 

Author's Reply to Reviewer 2

Specific Comments:

1) Section 2.4 Data Collection (starting at line 173) should include clear information about the interviewers’ personal characteristics and relationship with participants (COREQ #1-4, 6-8).

Reply: Thank you for your question. As it is known, many factors influence the chances of success in fieldwork with ethnic minorities, so the proposals to address the recruitment problems fall fundamentally within two areas of action in our study: entry strategies into the community and recruitment and sample diversification strategies. To overcome these barriers and increase the response rate, the researchers were from the same ethnic group - and sometimes the same gender - since it facilitates entry into the community and the relationship with the interviewees. In our study, all researchers received a training course about cross-cultural research and interviewing techniques before starting the recruitment. Participants were interviewed in their native language and by an interviewer from a similar ethnic group (see S2 Table for more information). However, there was no prior personal relationship with any of the participants. 

We have now specified this in the following paragraph:

Line 195: Participants were interviewed in their native language and by an interviewer from a similar ethnic group. BMM, MCA-B, YH, REH-E, FV and H conducted the interviews, and they had no prior personal relationship with any of the participants (see Table S2 for more information).

2) Line 194-195, the authors indicate interviews which were excluded from the analysis. However, if there was the occurrence of refusals or dropouts, the authors did not indicate as such (COREQ #13). Nor, was the presence of non-participants identified (COREQ #15).

Reply: Thank you for your comment. The dissemination of the study was performed as described in the prior comment, so those individuals that were interested in participating in the study contacted the study research team to be interviewed. As the participation was voluntary and the interview was performed in one session, the study did not have refusals nor drop-outs. Regarding COREQ #15, during the interviews there were only the participant and the interviewer, without any other people present. Following the COREQ guidelines we have added the necessary information to comply with COREQ #15 and #13, respectively:

Line 179: Due to the COVID-19 pandemic, interviews were administered by phone or using online video-call platforms ensuring data encryption for security and privacy reasons. Five interviews were led in person, as some participants did not know how to use information technologies. During the interviews there were only the participant and the interviewer, to avoid the influence of the presence of non-participants.

Line 201: As the participation was voluntary and the interview was performed in one session, the study did not have refusals or drop-outs.

3) Line 602-603, the last sentence is not necessary “Besides, it would be interesting to transfer the study findings to other stigmatized and oppressed groups in future studies.”

Reply: Thank you for your suggestion. Following it we have deleted the sentence.

Major Comments:

4) To better understand the diversity of the study population, the authors should comment on the reason of immigration (e.g. forced migration, economic, etc.). In a related capacity, the manuscript does not comment how this factor may impact the findings of the study. Were there any differential experiences based on this factor? Some explanation is required here.

Reply: Thank you for your comment. As we mentioned previously, our study aims to guarantee the diversification of the sample, reflecting the heterogeneity of the ethnic migrant groups in terms of discrimination in Spain. In our study, the reasons for migration were heterogeneous and depended on individual characteristics and no patterns were found between participants. For example, in the case of unaccompanied minor male migrants, they reported forced migration by poor living circumstances or voluntary migration, looking for a better future, depending on their personal situation. However, there is no apparent connection with discrimination experiences, and we neither did not directly ask this in the interview because it was not one of our main aims. 

5) In section 3.1 Distribution of Participants (line 221-222), the authors write “94 of 102 participants reported having experienced some type of discrimination since their arrival to the host country”. What were the experiences of the other 8? Was it revealed during the interview they in fact experienced some form of discrimination but did not perceive it as such? How were those interviews considered in the analysis and how might this influence the findings? Some explanation is required here.

 Reply: Thank you for your question, which has greatly enriched the article. For participants who did not report having been discriminated when asked directly, it was revealed that they had experienced some form of discrimination later in their narrative. However, it had not been perceived as such. The interviews were not treated differently, as even though there may not be a perception of discrimination, it does not mean that it does not affect the individual. Furthermore, in all cases, upon reading their narratives, it was evident that they had experienced discrimination.

Line 237: Most participants (94 of 102) reported having experienced some type of discrimination since their arrival in the host country. Those who did not report having been discriminated against when asked directly, later in their narrative, it was revealed that they had experienced some form of discrimination. However, it had not been perceived as such. This could be attributed to a lack of knowledge of the concept, not being aware of the phenomenon, or due to stigmatization, as some participants reported that for them, it meant having low self-esteem and a debility trait. An example of this experience was described by Participant 18 (P18): 

P18: I've never experienced it [discrimination], but similar things have happened to me, for example, there's people in the street with their phone and they are scared or walk away from the Moroccan [referring to himself], or you make a question to someone, and he/she doesn't answer.... [18 years, from Morocco]

---

## [Decision Letter · Decision Letter 1]

30 Oct 2023

Impact of Perceived Discrimination and Coping Strategies on Well-being and Mental Health in Newly-Arrived Migrants in Spain

PONE-D-23-16852R1

Dear Dr. Paula Cristóbal Narváez,

We’re pleased to inform you that your manuscript has been judged scientifically suitable for publication and is formally accepted for publication.

Kind regards,

AKM Alamgir, PhD

Academic Editor

PLOS ONE

Additional Editor Comments (optional):

Reviewers' comments:

Reviewer's Responses to Questions

**Comments to the Author**

1. If the authors have adequately addressed your comments raised in a previous round of review and you feel that this manuscript is now acceptable for publication, you may indicate that here to bypass the “Comments to the Author” section, enter your conflict of interest statement in the “Confidential to Editor” section, and submit your "Accept" recommendation.

Reviewer #1: All comments have been addressed

Reviewer #2: All comments have been addressed

2. Is the manuscript technically sound, and do the data support the conclusions?

Reviewer #1: Yes

Reviewer #2: Yes

3. Has the statistical analysis been performed appropriately and rigorously? 

Reviewer #1: Yes

Reviewer #2: N/A

4. Have the authors made all data underlying the findings in their manuscript fully available?

Reviewer #1: Yes

Reviewer #2: Yes

5. Is the manuscript presented in an intelligible fashion and written in standard English?

Reviewer #1: Yes

Reviewer #2: Yes

6. Review Comments to the Author

Reviewer #1: I am content with the revisions that the authors have implemented. I deem the article suitable for publication at this juncture.

Reviewer #2: No further comments from the first review. The authors of the manuscript accepted and/or adequately responded to the comments and feedback of both reviewers.

7. PLOS authors have the option to publish the peer review history of their article (what does this mean?). If published, this will include your full peer review and any attached files.

Reviewer #1: **Yes: **Kawalpreet Kaur

Reviewer #2: No

---

## [Editor Report · Acceptance letter]

13 Dec 2023

PONE-D-23-16852R1 

PLOS ONE

Dear Dr. Cristóbal Narváez, 

I'm pleased to inform you that your manuscript has been deemed suitable for publication in PLOS ONE. Congratulations! Your manuscript is now being handed over to our production team.

Kind regards, 

on behalf of

Dr AKM Alamgir 

Academic Editor

PLOS ONE